# Diversity and Assembly of Bacteria Community in Lime Soil under Different Karst Land-Use Types

Xiaoxiao Zou [1,2], Kai Yao [1,*], Fuping Zeng [2], Chen Zhang [2], Zhaoxia Zeng [2] and Hao Zhang [2,3,*]

1   School of Life Science, Guizhou Normal University, Guiyang 550025, China
2   Key Laboratory of Agro-Ecological Processes in Subtropical Region, Institute of Subtropical Agriculture, Chinese Academy of Sciences, Changsha 410125, China
3   State Key Laboratory of Environmental Geochemistry, Institute of Geochemistry, Chinese Academy of Sciences, Guiyang 550081, China
*   Correspondence: sc.catcher@hotmail.com (K.Y.); zhanghao@isa.ac.cn (H.Z.); Tel.: +86-180-8518-8442 (K.Y.); +86-186-8497-6587 (H.Z.)

**Abstract:** Bacteria play an important role as decomposers in karst ecosystems, which can be associated with karst soil and plants, promoting the cycling of nutrients between plants and soil. To reveal the diversity and structure of soil bacterial communities in some karst land-use types after the Grain for Green pattern, soil samples were collected from different land-use types (crops, grasses, and plantations) for analysis. Changes in the structure and assembly of karst soil bacteria were examined using 16S rRNA Illumina sequencing and soil chemical properties. We found that 18 years after the Grain for Green program, the soil chemical properties of available nitrogen (AN), total phosphorus (TP), total nitrogen (TN), and soil organic matter (SOM) of grassland were significantly different from those of farmland. The soil chemical properties in plantations were also significantly lower than those in farmlands. Different land-use types did not significantly affect the soil bacterial community structure. Actinobacteria, Proteobacteria, Chloroflexi, and Acidobacteria were the dominant phyla in all the plots. The relative abundance of Proteobacteria was higher in grasslands and plantations than in farmlands, whereas that of Actinobacteria was lower in grasslands. Furthermore, no significant correlations were observed between the soil chemical factors and soil bacterial groups at the genus level. The null model analysis indicated that dispersal limitations in stochastic processes predominated for the different land-use systems. Combined with previous analyses of the factors driving bacterial core species diversity in karst soils, we speculated that stochastic processes play a more important role in the construction of core bacterial species in restored karst soils at the plot scale.

**Keywords:** karst; soil bacteria; community diversity; null model; southwest China



## 1. Introduction

Soil microbial communities can regulate aboveground—underground biological cooperative regulation and efficient nutrient use. They are also strongly space-dependent and diverse, which influences soil and plant health [1,2]. Soil bacteria are the most abundant soil microorganisms that maintain biodiversity by regulating ecosystem processes and promoting nutrient cycling [3]. In terms of spatial distribution patterns on a global scale, altitude span, temperature, and precipitation were more closely associated with the distribution of soil microorganisms in mountain soils, indicating that climate is the main factor involved in the large-scale geographical distribution patterns of soil bacteria. Vegetation and soil elements are secondary contributors to climate gradients [4,5]. At the regional scale, varying ecological environments such as climate, vegetation, and soil, caused by differences in altitude gradients, lead to different geographical distribution patterns of soil bacteria along altitude gradients. However, the existing studies have many limitations [6]. At the meso- or microscale, studies of soil bacterial diversity are limited by the physical and chemical properties of the soil, fauna, and rhizosphere microhabitats [7]. Therefore,

studying the distribution patterns and driving factors of soil bacterial community diversity is of great significance for exploring the organic combinations of aboveground and underground processes and soil functions.

Previous studies have gradually deepened the exploration of the diversity of soil bacterial communities in different habitats and the internal community structure mechanisms. Community assembly determines the composition and diversity of bacterial communities, which further affects their ecological function. Soil bacterial community diversity is largely caused by deterministic factors such as niche differentiation, or random factors such as ecological drift [8]. The balance among these factors is regulated by environmental conditions [9]. Deterministic processes are determined by homogeneous or heterogeneous selection, whereas stochastic processes are determined by diffusion (homogeneous or stochastic diffusion) or drift. Many studies have quantified the relative importance of stochastic versus deterministic processes that drive microbial community construction in different habitats and at different scales (such as regional, continental, and global scales); however, the results remain controversial. Previous studies have found that soil bacterial community assembly is dominated by deterministic processes at arid, cold, and high altitudes, whereas it is dominated by stochastic processes at warmer and lower altitudes [10]. Current research includes the temporal and spatial changes in bacterial communities in natural restoration vegetation and their influencing factors; however, changes in bacterial community structure in artificially restored types remain unclear. Therefore, understanding the diversity and assembly mechanisms of soil bacterial communities during different land-use systems can provide a theoretical basis for restoring degraded soils and their ecological functions.

Karst is a special terrain or landscape that grows on carbonate rocks such as limestone and dolomite [11,12], accounting for approximately 15% of the global land. Karst areas in southwest China are among the most fragile ecosystems in the world and are threatened by exposure to volatile conditions, serious human interference in growth and development, and extreme rocky desertification [13]. Rocky desertification causes serious soil and water loss, bare leakage in large bedrock areas, a sharp decline in productivity, and the emergence of a rocky-desertification landscape, all of which seriously hinder social and economic development. To combat rocky desertification and alleviate poverty, the Chinese government launched the nationwide Grain for Green pattern (GfG) in 1999 to return farmland to plantations and grasslands. Over the past two decades, this project has restored most of the degraded land in the region through natural enclosures, afforestation, and pasture planting [14,15]. In the process of karst ecological restoration, the three elements in the plant-soil-microbial symbiotic system have a "co-succession" effect [16,17]. Studies have found that the microbial community composition in karst regions is similar at different succession stages, and that soil pH and phosphorus are the main factors determining community structure [18,19]. However, some studies have found that different vegetation successions significantly affect the soil bacterial diversity in karst areas [20]. Although previous studies have investigated soil microbial diversity and community composition in karst natural vegetation restoration, the diversity and community composition of soil bacterial communities in the artificial restoration of karst land-use types remain unclear.

In this study, we measured the effects of different restoration patterns on soil chemical properties and soil bacterial composition using 16S rRNA sequencing technology in three different land-use types: crop (maize), grassland (Guimu-1 elephant grass), and plantations (*Swida wilsoniana*). This study aimed to explore the response of soil bacterial diversity to land-use types and the factors that influence this response. Furthermore, we explored the mechanism of soil bacterial community formation on karst slopes using a null model and provided a theoretical basis for soil health management of karst land-use types in southwest China.

## 2. Materials and Methods

### 2.1. Research Site

The study area was situated in the Huanjiang Karst Ecosystem Observation and Research Station, Chinese Academy of Sciences, Huanjiang Maonan Autonomous County, Guangxi Zhuang Autonomous Region (24°43′ N–24°45′ N, 108°18′ E–108°20′ E) (Figure 1). The area has a subtropical monsoon climate, with an average annual temperature of 19.9 °C and an average temperature of 29 °C in July. The average annual rainfall is 1400–1500 mm, and the rainfall from April to August accounts for 70% of the annual rainfall.

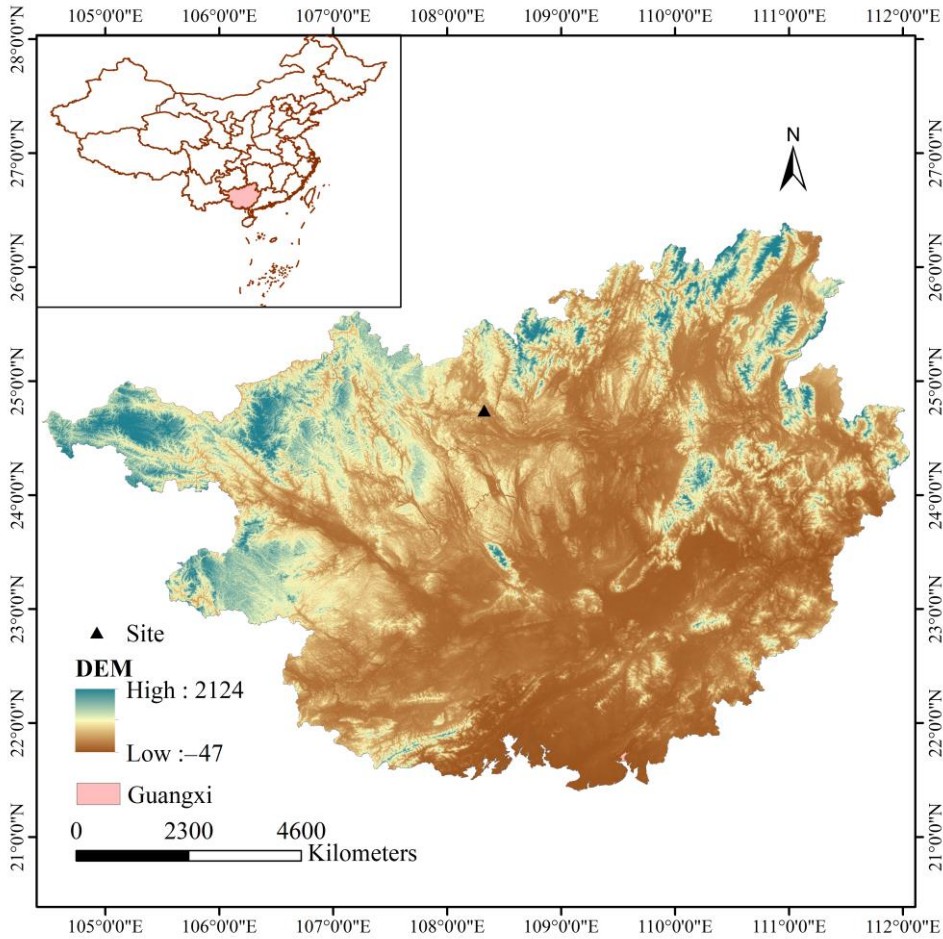

**Figure 1.** The research sites in this study.

In 2004, farmland with relatively uniform soil and vegetation on the slope was selected to establish an observation platform for reclamation along the slope to simulate different ecological restoration methods. A long-term controlled experimental plot with a similar area and slope was established. Each plot on the long-term monitoring platform was 20 m wide and 70 m long, with a slope < 25°. Along the slope, the average soil thickness increased from 10–30 cm to 50–80 cm. The developed soil was mainly brown lime soil and the surface was mostly covered with gravel (particle size > 2 mm), which content reached 10%–40% in the topsoil. The contents of sand and clay were 25%–50% and 30%–60%, respectively [21]. The sample plot was surrounded by a brick-and-mortar enclosure (approximately 20 cm on the surface and underground), which was mainly used to block the surface runoff and sediment in and out of the area. After establishing the observation site, maize (*Zea mays* Linn.) was monocultured twice a year, plowed before sowed, and chemical fertilizer was used in the farmland area. The plantation was an artificial *Swida wilsoniana* forest and the grassland was an artificial Guimu-1 elephant grass. Guimu-1 elephant grass is harvested 3–5 times a year and replanted every 5 years. In the *Swida wilsoniana* plantation, there is high

canopy density and a small amount of herbaceous (10–20 cm) and shrubs (20–50 cm). After the establishment of the plot, the plantation did not receive any fertilization or artificial management and, before that, the various areas had the same land-use and management practices as crop and grassland.

## 2.2. Determination of Soil Chemical Properties

In July 2022, three types of land-use were selected: crop (maize), grassland (Guimu-1 elephant grass), and plantation (*Swida wilsoniana*). The sample size was changed to 60 m × 20 m, excluding the 10 m distance near the road with a relatively serious human disturbance at the bottom. The plot was divided into 12 quadrats (10 × 10 m). Sampling points in the woodland were at least 0.5 m from the tree trunks, and impurities and organic layers were removed from the soil surface before sampling. In Guimu-1 elephant grass and farmland, the location between the two plants was selected for sampling, avoiding the selection of rhizosphere soil samples. The top 0–20 cm of the surface soil layer was collected using a soil auger (inner diameter, 5 cm). Each quadrat was sampled according to a random five-point sampling method and then all 5 samples were pooled into one sample, which was used as the composite quadrat sample. Twelve mixed soil samples were obtained for each land-use pattern, and 36 soil samples were collected. The stones and roots in the soil samples were removed immediately after sample collection, and the soil samples were filtered through a 10-mesh sieve and divided into two parts. One portion was placed in a ventilated room, and the sample was air-dried naturally for soil chemical properties measurement. The other sample was stored at −80 °C for high-throughput DNA sequence analysis after soil bacterial DNA extraction. The soil chemical properties were measured using the method described by Bao et al. [22], which include soil organic matter (SOC), total nitrogen (TN), available nitrogen (AN), available phosphorus (AP), total phosphorus (TP), and pH.

## 2.3. High-Throughput DNA Sequencing and Bioinformatics Analysis

Soil bacterial DNA was extracted from 0.5 g fresh soil using the Fast DNA SPIN kit for Soil (MP). The DNA concentration and purity were determined using a Nanodrop2000 ultra microphotometer (Nanodrop Technologies, Wilmington, DE, USA). DNA integrity was examined by electrophoresis on a 1.0% agarose gel, and the DNA was stored at −80 °C. The primers used for the bacterial 16S rRNA gene belong to the V3-V4 region. They were 338 F (5′-ACTCCTACGGGAGGCAGCAG-3′) and 806 R (5′-GGACTACHVGGGTWTCTAAT-3′). PCR was performed in a 20 μL reaction system (TransStart Fastpfu DNA Polymerase). PCR products from the same sample were pooled in triplicates and electrophoresed on a 2% agarose gel. The PCR products were cut from the gel and recovered using an AxyPreDNA Gel Recovery Kit (AXYGEN, Union City, CA, USA). The extracted DNA was sent to Shanghai Meiji Biotechnology Co., Ltd. (Shanghai, China) for sequencing on an Illumina MiSeq PE300 platform.

The PE reads obtained using MiSeq sequencing were spliced using FLASH, and the sequence quality was controlled and filtered using Fastp. Chimeric sequences were removed before further analysis. Representative operational taxonomic units (OTUs) at a 97% similarity level were clustered using the RDA classifier Bayesian algorithm in the Uparse 11 software. Representative bacterial sequences were separately aligned against the Silva database for species annotation and sequences for "chloroplast" and "mitochondria" were removed. The obtained OTUs were homogenized using the minimum sample sequence method, and subsequent analyses of diversity were based on the standardized abundance of OTUs.

## 2.4. Statistical Analysis

To determine significant differences in the soil's chemical properties, a one-way analysis of variance (ANOVA) followed by Tukey's test was performed using R4.2.1. Taxonomic analysis of the representative OTUs was performed using the RDP classifier Bayesian

algorithm. The species composition and relative abundance of each sample were calculated, and bar charts and Venn diagrams were generated. Mothur was used to calculate the α-diversity of soil bacterial community using Sobs, Shannon–Wiener, Simpson, and Chao indices. QIIME was used to calculate the beta diversity distance matrix for the hierarchical clustering analysis. The sample distance heatmap was generated using the Bray–Curtis algorithm. Species with significant differences between different land-use types were analyzed using the Linear discriminant analysis Effect Size (LEfSe) analysis. The vegan package in R was used to generate a box plot and principal component analysis (RDA) community heat map.

Null model analysis was performed to assess the assembly of bacterial communities. In the model construction, the β-nearest taxon index (βNTI) and community-based Raup–Crick (RCbray) matrix, constructed using the standard Bray–Curtis matrix, were measured to quantify the relative importance of deterministic and stochastic processes. When /βNTI/ > 2, community turnover is controlled by variables or homogeneous selection. When /βNTI/ < 2 and /RCbray/ > 0.95, community turnover is affected by uniform dispersal or dispersal limitation, while /βNTI/ < 2 and /RCbray/ < 0.95 indicate the influence of non-dominant processes. The relationship between the βNTI values and the Euclidean distance matrix of soil variables was assessed, and the Mantel test was performed simultaneously to assess changes in community assembly processes under different land-use types.

## 3. Results

### 3.1. Soil Chemical Properties in Different Karst Land-Use Types

The soil chemical properties under different land-use types were significantly different ($p < 0.05$), except for the pH (Figure 2). The contents of TN, AN, SOM, and TP in the three points were the highest in grassland, followed by farmland and plantation. Available phosphorus was the highest in farmland, followed by grassland and plantation. Our study area is a Grain for Green plot, with alkaline soil (7.59–8.00). Except for a significant difference between the soil pH values of grasslands and plantations, there was no significant difference between the pH values.

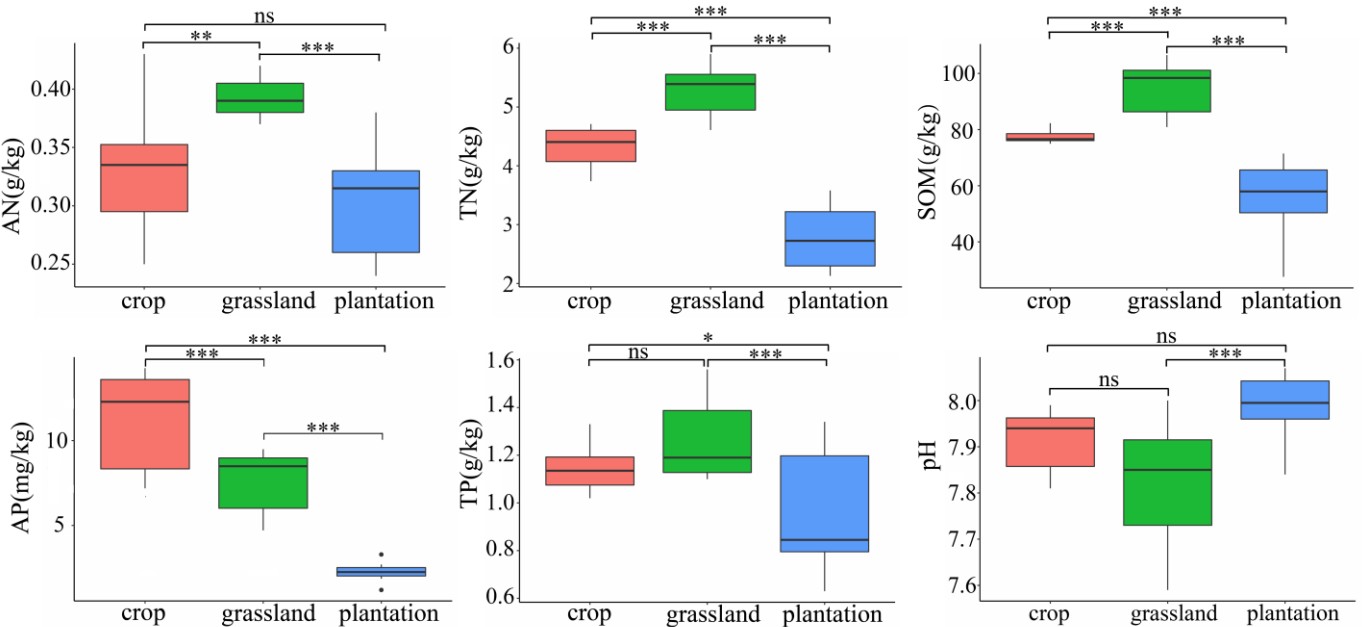

**Figure 2.** Different soil chemical properties in three karst land-use types. ***, $p < 0.001$; **, $p < 0.01$; *, $p < 0.05$; ns, $p > 0.05$.

### 3.2. Soil Bacterial Community Composition in Different Karst Land-Use Types

A total of 1,423,950 valid sequences with an average length of 419 bp were obtained from all the soil samples using high-throughput sequencing analysis. A total of 4548 sequences were classified as OTU and further classified into 36 phyla, 111 classes, 249 orders, 383 families, 667 genera, and 1363 species (Figure 3). The number of OTU species varied among the different land-use types: 3952 in farmland, 3698 in pasture, and 3868 in plantation. A total of 3009 OTUs were common to the three plots, whereas 262 (5.76%), 128 (2.81%), and 197 (4.33%) OTUs were unique to the cropland, pasture, and plantation species, respectively.

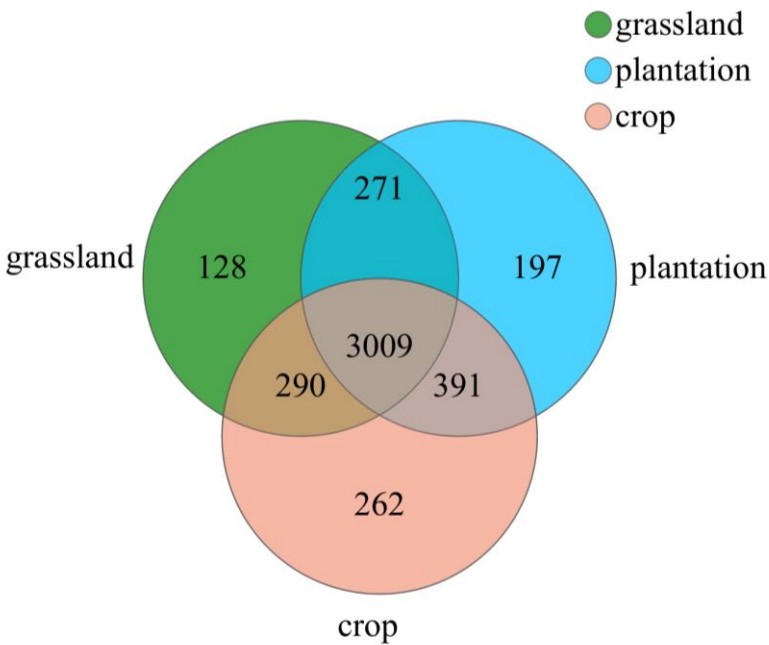

**Figure 3.** OTU Venn map of soil bacteria in three karst land-use types.

At the phylum level, the relative abundances of soil bacterial groups differed among the three land-use types. Actinobacteriota, Proteobacteria, Acidobacteriota, Chloroflexi, and Methylomirabilota accounted for 80% of the total bacteria (Figure 4). Actinobacteria was the most dominant, with a relative abundance of 31% in the three land-use systems, followed by Proteobacteria (18.31%), Acidobacteria (17.73%), Chloroflexi (8.13%), and Methymonas (5.50%).

At the genus level, six dominant taxa were observed in the three land-use groups, namely norank_f_norank_o_Gaiellales, norank_f_norank_o_Vicinamibacterales, norank_f_Vicinamibacteraceae, norank_f_67-14, norank_f_norank_o_Rokubacrteriales, and Gaiella (unclassified scientific names in taxonomic lineages were labeled as unclassified; if no scientific name is available in the classification system, the taxon is labeled as "norank") (Figure 5).

Figure 6 shows a heat map of the top 50 genera, indicating that the soil bacterial communities in the three land-use types can be divided into two groups with unique compositions and abundance. Croplands and grasslands had similar bacterial community compositions. Additionally, *norank_f_norank_o_Gaiellales*, *norank_f_norank_o_Vicinamibacterales*, and *norank_f_67-14* were highly abundant in all three land-use types. The relative abundance of *norank_f_norank_o_norank_c_Subgroup_18* was low in plantations and grasslands, whereas that of *Bryobacter* was low.

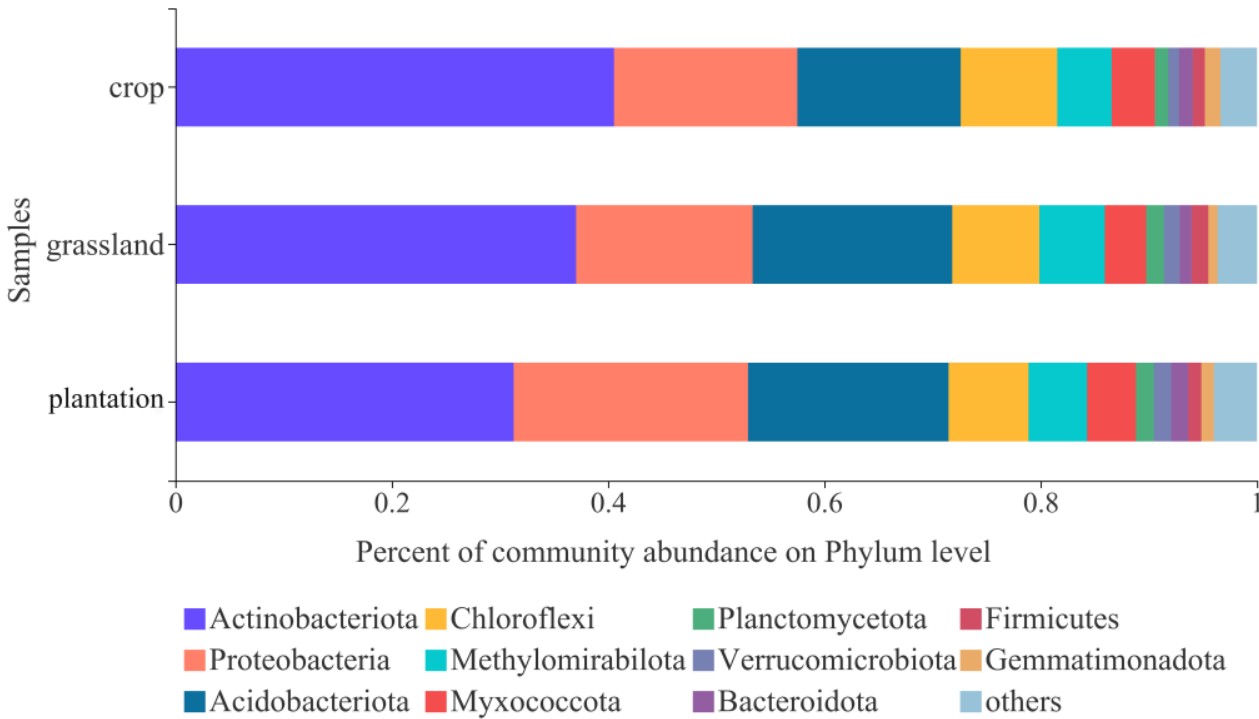

**Figure 4.** Relative abundances of soil bacteria at phyla level in three karst land-use types.

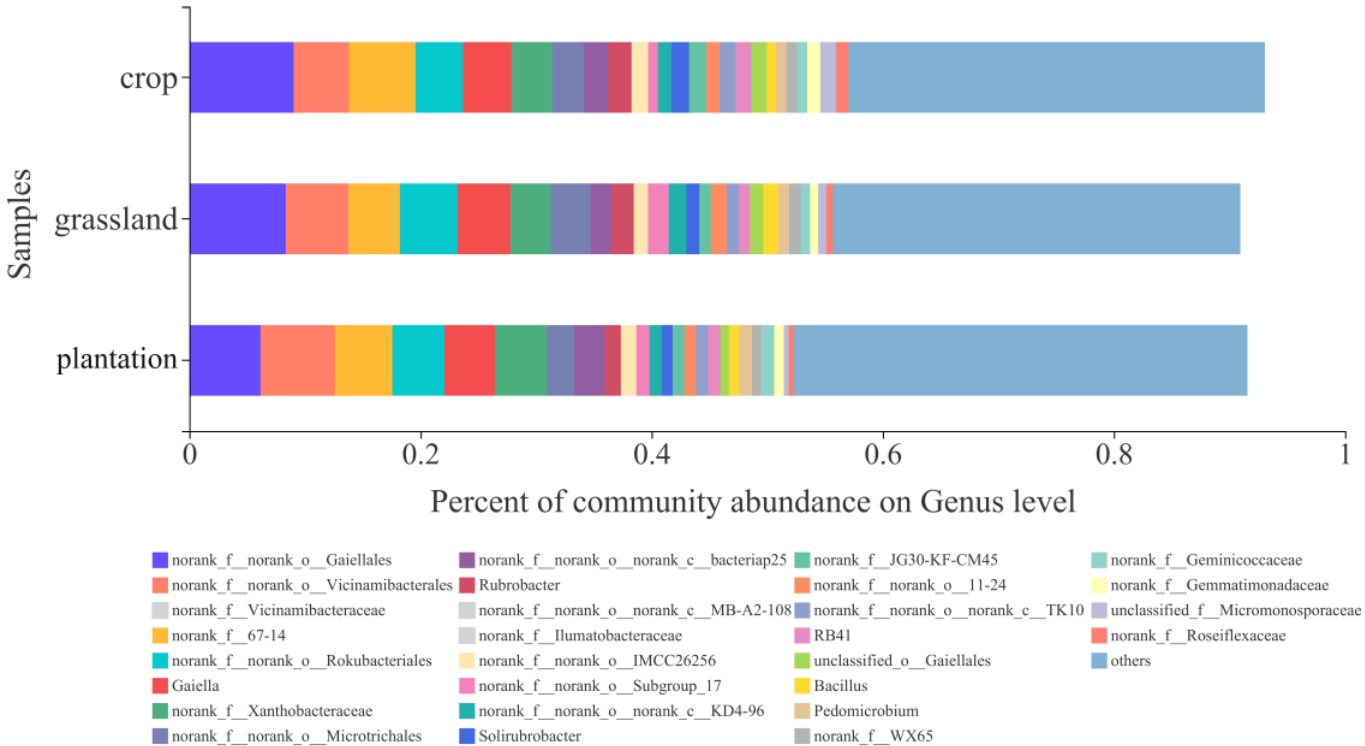

**Figure 5.** Relative abundance of soil bacteria at genera level in three karst land-use types.

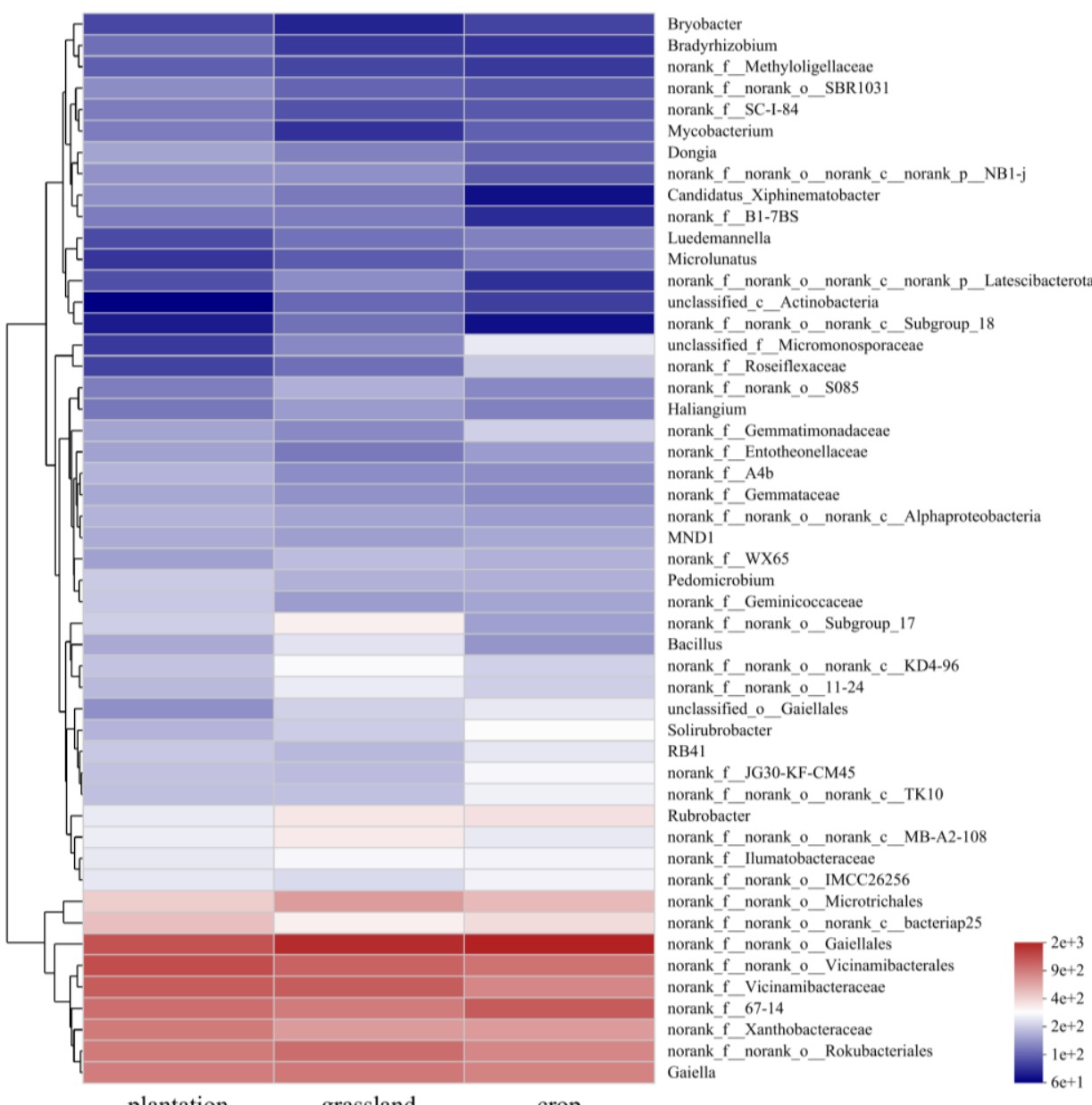

**Figure 6.** Heat maps and hierarchical clustering of the top 50 genera of soil bacterial community abundance in three karst land-use types.

No significant differences in the richness and diversity indices were found among the three plots (Figure 7). Chao, Simpson, Sobs, and Shannon diversity indices were the highest in farmland, followed by grassland and plantation.

LEfSe analysis of the soil bacterial communities at the genus level in the three land-use types revealed six different genera (Figure 8): *Actinobacteria* and *Micromonosporales* varied widely in farmland; *Subgroup_17* in grassland; and *Cellvibrionales*, *Corynebacteriales*, and *Caulobacteriales* in plantations.

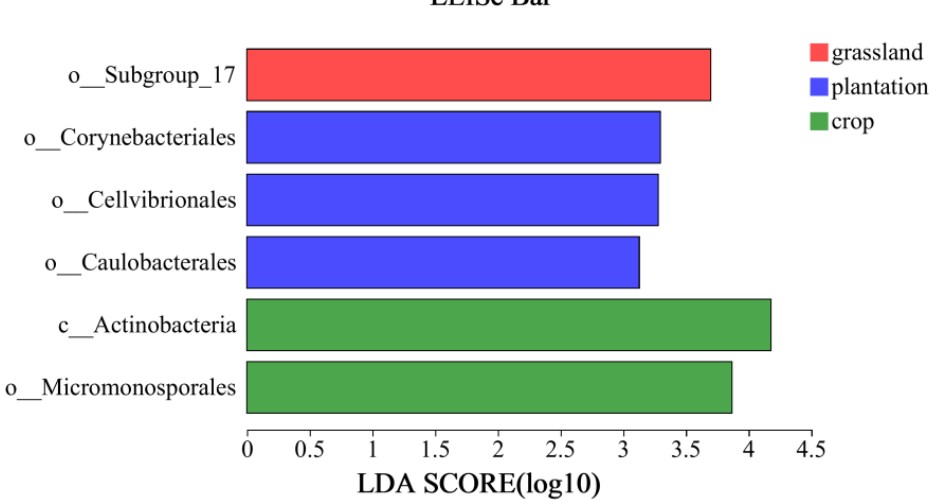

**Figure 7.** Alpha diversity index of soil bacterial community in three karst land-use types. ns, *p* > 0.05.

## LEfSe Bar

**Figure 8.** Histogram of LDA value distribution of soil bacterial species in three karst land-use types.

### 3.3. Association between Soil Chemical Properties and Soil Bacteria in Different Karst Land-Use Types

No significant correlation was observed between the soil chemical properties and soil bacteria at the genus level (Supplementary Figure S1). TN, AP, and pH were slightly correlated with the relative abundance of soil bacteria, whereas the other soil chemical factors had weaker correlations. Some soil bacterial communities were insensitive to soil physicochemical properties, and all physicochemical properties had a significant effect on the abundance of *unclassified _f_Micromonosporaceae*.

Redundancy analysis (RDA) between soil bacterial communities and environmental factors showed that the soil's physical and chemical properties could explain 20.08% of the variation in the soil bacterial community at the OTU level (Figure 9). The first and second axes explained 12.42% and 7.66% of all information, respectively. On the first axis, TN ($p = 0.001$, $R^2 = 0.2484$), AN ($p = 0.001$, $R^2 = 0.4555$), and AP ($p = 0.001$, $R^2 = 0.4088$) were significantly positively correlated with bacteria at the OTU level, indicating that soil nitrogen and phosphorus had greater effects on the distribution of bacterial communities. Furthermore, it is the dominant factor affecting the structure and distribution of bacterial communities.

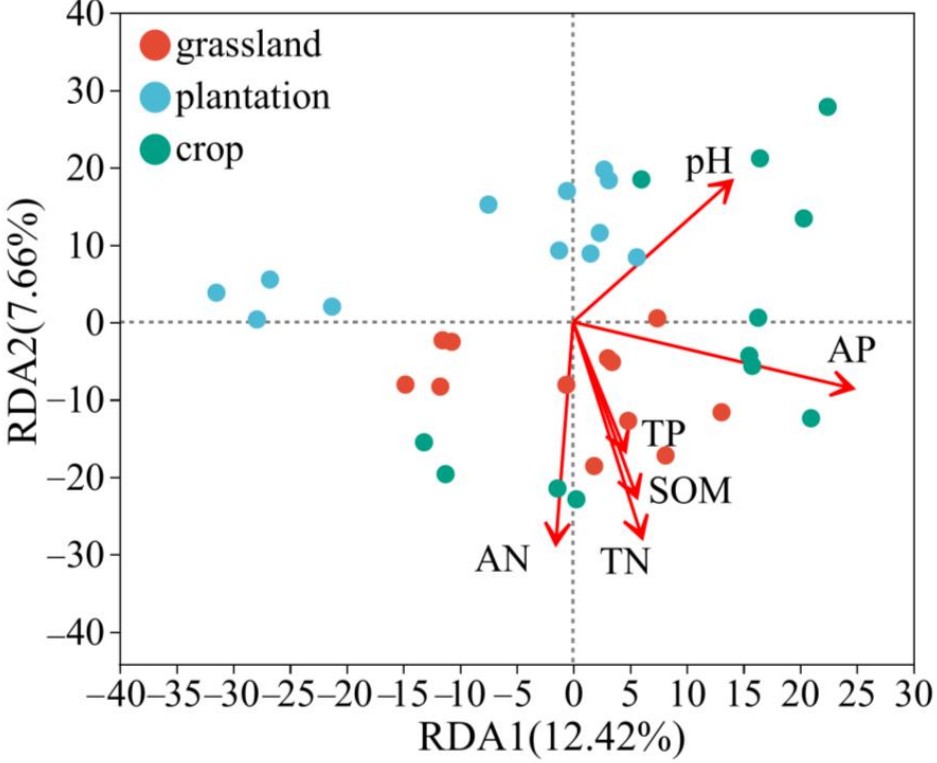

**Figure 9.** RDA analysis of bacterial community at OUT level and soil physicochemical factors in three karst land-use types.

### 3.4. Assembly Mechanism of Soil Bacterial Community in Different Karst Land-Use Types

To explore the differences in the assembly mechanism of different soil microbial communities, we calculated the βNTI and the RCbray based on null model analysis of taxonomic phylogenetic diversity to evaluate the ecological process of community construction. The results (/βNTI/ < 2, RCbray > 0.95) indicated that the diffusion limitation of stochastic processes dominated the bacterial community assembly in the three land-use types (Figure 10).

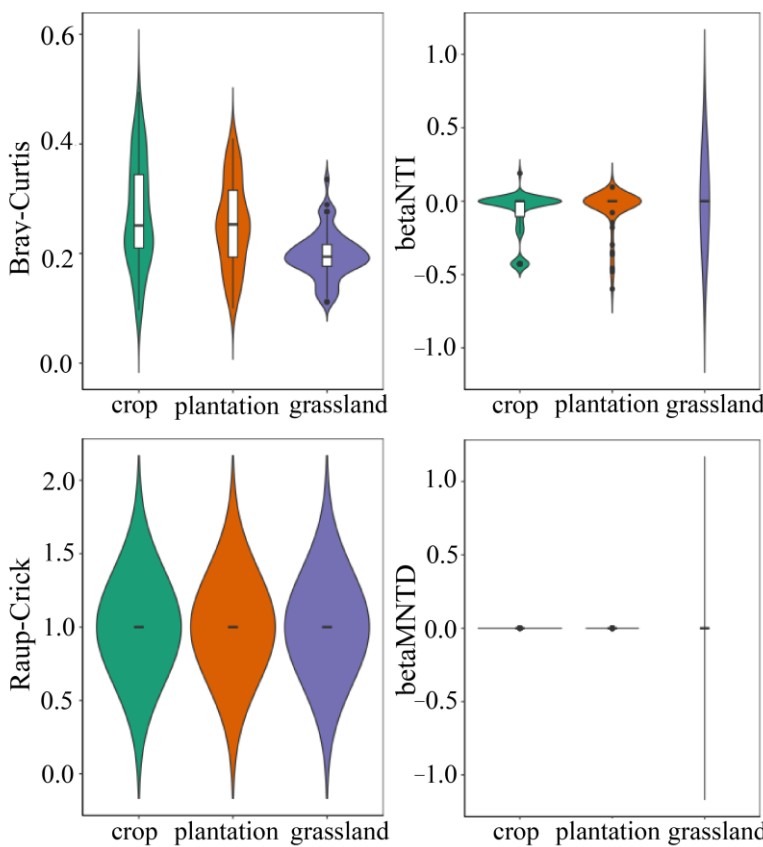

**Figure 10.** Null model of soil bacterial community in three karst land-use types.

## 4. Discussion

### 4.1. Changes in Soil Chemical Properties in Different Karst Land-Use Types

We found that soil properties under different treatments were significantly different. When the use of fertilizer was stopped after plantation, the soil nutrient content, particularly SOM, TN, AN, and AP, was significantly reduced (Figure 2). However, because of the short growth cycle, developed root system, and good water and fertilizer retention ability of Guimu-1 elephant grass in grasslands, the nutrients used by the grass can be quickly supplemented momentarily, resulting in a gradually higher nutrient content than that of farmland. Afforestation has been found to improve soil nutrient levels in the Loess Plateau region [23]. However, in the present study, the soil nutrient content did not improve, which may be related to the thin soil layer and low nutrient content in the karst area; thus, it is difficult to replenish the nutrients in a short time after plants absorb them. The soil pH did not change significantly after afforestation, which is consistent with previous studies [19,24], indicating that different land-use types had little effect on the pH of karst soil. In general, compared to the physical and chemical properties of cultivated land, the improvement of soil by plantation was not as evident as that of grassland soil, indicating that grassland may contribute to the restoration of degraded land, whereas the restoration of soil properties by plantation was relatively long owing to its long growth cycle. Lan et al. [25] found that the soil chemical properties of restored farmland in karst areas after 20 years of natural restoration were significantly lower than those of natural forests, and the restoration to the pre-reclamation state was difficult. However, the protective effect of restored farmland on surface soil lasted extensively, and the continuous deterioration of soil chemical properties could be controlled effectively. Furthermore, Chen et al. [19] found that the combination of *Zenia insignis* and Guimu-1 elephant grass improved soil chemical properties better than individual treatments. Therefore, planting grass in artificial forests may increase plant diversity and accelerate the recovery of the physical and chemical properties of soil.

### 4.2. Changes in Soil Bacterial Community Diversity and Composition in Different Karst Land-Use Types

Compared to farmland, the $\alpha$-diversity of the soil bacterial community in the Grain for Green program showed a decreasing trend (Figures 3 and 7), but the difference was not significant. This observation indicated that the Grain for Green program could not significantly restore the soil bacterial community. Our findings are similar to those of previous studies [26]. No significant differences among the different land-use types were observed, indicating that the microbial community structure in a certain area is more affected by the soil type [27]. This shows that, after the degradation of the soil ecosystem, restoration to the ideal state is difficult, and it may recover slowly compared with aboveground vegetation [28]. Buckley [29] and Jangid [30] also showed that the soil bacterial community was not sensitive to changes in vegetation community, and vegetation mainly improved soil properties through vegetation litter and root exudates, which lasted for a long time.

At the phylum level, the conversion of the Grain for Green to grassland did not change the dominant phyla of soil bacterial communities; however, their relative abundance significantly changed. The dominant bacterial phyla, including Actinobacteria, Proteobacteria, Acidobacteria, Chloroflexi, and Methymonas (Figure 4), are consistent with the results of previous studies [31–33]. Actinomycetes are associated with cellulose and lignin decomposition and may be related to the recently implemented policy of returning straw to the field, where maize straw is artificially crushed, resulting in the highest relative abundance of actinomycetes in farmlands to accelerate litter decomposition [34]. Previous studies have utilized high-throughput technology to study the soil bacterial community structure in evergreen and deciduous broadleaved forests in Shenlongjia and found that high organic matter content is related to a high relative abundance of Actinobacteria [35], which is consistent with the results of our study. Proteobacteria are eutrophic bacteria that rapidly multiply in nutrient-rich soils [36]. However, in the present study, the relatively high abundance of Proteobacteria in plantations with relatively low-nutrient content suggests that other factors influence Proteobacteria. This may be related to the fact that Proteobacteria contain many subpopulations from different habitats and have a wide ecological niche. Meanwhile, the relative abundance of Acidobacteria was slightly lower in cropland than in plantation and grassland (Figure 4), which is consistent with previous studies showing that Acidobacteria are more abundant in nutrient-poor environments [32,33,36]. Chloroflexi have a strong tolerance to stress and are suitable for reproduction in nutrient-poor soil environments. However, the relative abundance of Chloroflexida was slightly higher in farmlands in this study (Figure 4), which may be related to the use of organochlorine pesticides. Various studies have shown that the application of organochlorines increases Chloroflexida due to the inherent ability of organochlorine digestion [37].

*Bacillus* is a phosphate-soluble bacterium, which can convert insoluble phosphate in soil into phosphate that can be absorbed by plants [38]. The relative abundance of *Bacillus* was high in the three plots and was significantly positively correlated with TP content, which was of great significance for improving phosphorus (P) utilization rate, plant P nutritional status, and soil conditions. The relative abundance of *Gaiellales* in crop is higher than in plantation and grassland; *Gaiellales* are aerobic bacteria, and crop tillage increases soil oxygen content. Kraut-Cohen et.al. [39] studies found that the content of *Gaiellals* in tillage farmland is significantly higher than in untillage farmland.

### 4.3. Effects of Soil Properties on Soil Microorganisms in Karst Land-Use Types

Changes in the microbiome and environmental factors are often co-variant. Many environmental variables can explain microbial community composition [40,41]. In the present study, RDA analysis showed that the distribution of bacterial diversity was mainly affected by TN, AN, and AP (Figure 8), indicating that the heavy use of chemical fertilizers in farmland not only accumulated soil nutrient content but also affected bacterial community structure. Karst soils in subtropical climates are characterized by N and P deficiencies; therefore, low N and P concentrations were the most important limiting factors. Many

studies have shown that the soil's pH is a key factor affecting bacterial communities [42,43], which is inconsistent with the results of the present study, and may be due to the non-significant correlation between pH changes [30]. Li et al. [44] found that the pH of shallow soil in the Loess Plateau was not related to the bacterial community, which was consistent with our results.

In this study, there was no significant relationship between soil chemical factors and bacterial communities (Figure S1). However, the soil bacterium Gaiellales was significantly positively correlated with the soil AP, TP, and SOM. This taxon belongs to Actinobacteria, which has diverse nutrient patterns and metabolic pathways and is actively involved in the cycling of carbon, nitrogen, and other elements in the soil [45]. Furthermore, the soil indicators in the RDA analysis explained only 20.08% of the variation (Figure 8), suggesting that the soil bacterial community in the study area may be affected by other environmental factors. Therefore, the effects of vegetation, soil, topography, and climate on soil bacterial communities should be comprehensively analyzed in the future.

### 4.4. Community Assembly of Soil Bacteria in Three Karst Land-Use Types

Our results suggest that the restoration of artificial vegetation may lead to a random influx of microorganisms. The dominance of stochastic processes can generate more diverse ecological functions to maintain the stability and persistence of ecosystem functions, which are consistent with those of previous studies [46–48]. Additionally, the influence of the diffusion limitation on the stochastic process was found to be greater than that of homogeneous diffusion and uncertain processes (Figure 9). Bacterial communities tend to be less similar because of diffusion limitations, which may be affected by soil substrate diffusion resistance [48,49]. Kong et al. [50] found that afforestation in the Loess Plateau shifted the soil bacterial community composition from homogeneous to random diffusion, which may be related to changes in water and soil nutrients. Jiao found that long-term irrigation, fertilization, and crop cultivation promoted homogeneous diffusion in farmland, which led to a convergence in bacterial community composition [51]. However, in this study, the process of community construction in different land-use types was a random diffusion, indicating that the process was not related to soil nutrients. Vegetation reconstruction is key to the transition of diazotrophic flora from deterministic to stochastic processes in degraded karst ecosystems [52].

Deterministic and stochastic processes jointly drive community structure in different environmental contexts, but with different importance to each other [53,54]. Previous studies have found that soil microbial community structure is controlled by stochastic processes in the early stages of vegetation succession, whereas the influence of deterministic processes gradually increases in the later stages [55–57]. In the Loess Plateau region, deterministic processes have always dominated the construction process of soil bacterial community structures in coniferous forests under various habitats [58]. However, in the process of vegetation restoration in karst areas, homogeneous diffusion in the stochastic process dominates the construction of soil fungal communities [59]. Additionally, bacterial and fungal interactions are common in soil and play important roles in many ecological processes. However, in this study, we analyzed bacterial community diversity and construction in different land-use types and did not include fungal community diversity and construction; therefore, the future scope of this study will include the effects of different land-use types on soil fungal community diversity and construction. The relationship between biodiversity and ecosystem function (BEF) is an important ecological research area. In recent years, the relationship between plant diversity and ecosystem functions has been studied, and it has been found that there is a positive correlation between plant diversity and productivity. Many studies have gradually progressed from species diversity to functional diversity and beta diversity, and from a single function to ecosystem versatility. However, the relationship between karst microbial diversity and versatility of degraded ecosystems remains unclear. Further studies on the relationship between karst microbial diversity and ecosystem versatility are required.

## 5. Conclusions

In this study, we examined soil bacterial communities under different land-use types in karst areas of southwest China. After 18 years of Grain for Green pattern, we found that different land-use types did not affect soil bacterial community diversity and community structure, but soil chemical properties were significantly different under different land-use types. The results showed that soil chemical properties in grassland were significantly higher than those in cropland and plantation. The dominant bacterial phyla did not change much from farmland to plantation. There was no significant correlation between soil chemical factors and soil bacterial genus levels. Dispersal limitation was the main mechanism of bacterial community construction. This study provides biological theoretical support for ecosystem maintenance in other fragile ecosystems around the world. Future studies should focus on the effects of different land-use types on the diversity and construction mechanisms of soil fungal communities.

**Supplementary Materials:** The following supporting information can be downloaded at https://www.mdpi.com/article/10.3390/f14040672/s1, Figure S1: Correlation between major bacterial communities in soil and soil chemical factors in three karst land-use types.

**Author Contributions:** Conceptualization, H.Z. and X.Z.; funding acquisition, F.Z., H.Z. and K.Y.; investigation, X.Z., K.Y., C.Z. and Z.Z.; methodology, F.Z.; writing—original draft, X.Z. and H.Z. All authors have read and agreed to the published version of the manuscript.

**Funding:** This research was supported by the National Key Research and Development Program (2022YFD1300805), the National Natural Science Foundation of China (31870712, 32071846, 42071073), the Natural Science Foundation of Hunan Province (2021JJ30764), the State Key Laboratory of Environmental Geochemistry (SKLEG2021207), and by the Hechi City Program of Distinguished Experts in China.

**Data Availability Statement:** Not applicable.

**Conflicts of Interest:** The authors declare no conflict of interest.

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
