# Peer review of "Diversity and Assembly of Bacteria Community in Lime Soil under Different Karst Land-Use Types"

_forests, doi:10.3390/f14040672_

Round 1
Reviewer 1 Report
In the manuscript "Community diversity and assembly of soil bacteria in karst plantation vegetation" soil bacteria from three different types of vegetation are analyzed 18 years after planting. These were extremely poor and exploited karst soils which they tried to recover through an artificially restoration of the vegetation.
The work is placed in an interesting framework of microbial ecology. It is well written and addresses the issues correctly. In fact, the physicochemical and microbiological analyzes of the soil give results that are not very relevant, compared to what could have been expected, despite the 18 years in which the vegetation had the opportunity to develop.
Of the three types of vegetation, the one that gives less consistent results is that of Swida wilsoniana. This amazed me and I wondered if there is any grass in the undergrowth. After 18 years grass should be there. How tall are the plants? If there is grassing there should be plant biodiversity which stimulates microbial biodiversity and better nutrient cycling. In particular, phosphate solubilising microorganisms appear to be completely absent. In a karst soil with a high pH, all phosphorus is made unavailable to plants, unless id solubilized by microorganisms. A final factor that may have had a negative influence is the absence of mycorrhizae.
A final consideration, which you also make, is that in the absence biodiversity at the plant level, it is difficult for stable and efficient ecosystems to form and for the bacterial community to be biodiversified.
In particular:
Lines 95-96. You should explain better what you mean by “farmland (maize), grain for grass (pasture), and grain for forest (plantation).” A land sown for 18 consecutive years with maize? It seems very strange to me. Has the land been worked/plowed every year? Perhaps sod sowing was done?
As for the elephant grass, has it been standing for 18 years or has it been grazed? Or was it mowed down?
As for Swilda, what does “grain for forest” mean? Also, it is fine to call it "plantation" but not "forest". By forest we mean a biodiversified natural ecosystem with trees, shrubs, grasses of different species. You must remove this definition throughout the manuscript.
On lines 143-146. Move the entire words from lines 201-202. For the first time, do not use acronyms here. NaHCO3 the 3 goes in subscript.
Moving Figure 2 from Materials and Methods to Results.
In the captions of figures 4 and 5 it cannot be said “Species composition of Phyla and Genera”. Rephrase.
Line 293. “the soil pH”. “The” is capitalized.
Reviewer 2 Report
The manuscript requires significant additions and changes before publication. Below are the comments.
Line 114-116 - it says that the soils contained 10 to 40% gravel. The content of sand, dust and clay was not specified. This is necessary for further inference. Some chemical properties of soils were determined in the research. Their differentiation could be due to the different content of dust and clay, and not the way of exploitation. Therefore, listing of the dust and clay content is necessary. In the Materials and Methods chapter, there is no information about which region of 16SRNA was studied. In the same chapter it is written that soil chemical determinations were carried out in the same way as described in ref. 19. This is erroneous, as item 19 does not describe these methods, just another reference to the literature. It seems that the discussion of the results should be more chronological. Please start the description with phylum and then move on to the next taxa, i.e. classes, orders, families and genera. Line 215 - the authors refer to Figure 3, which shows the Venn map of soil bacteria in karst plantation vegetation, and in the text they describe how many types, classes, etc. they have classified. In addition, it is puzzling to note that the authors have classified 1,363 species. Please specify the caption in Figures 4 and 5. In their current form, they are not compatible with the information on the number of taxa classified. If only those that occur above a certain percentage in the soil are presented, please write it under the figure or in the methodology. Now it's unclear. In addition, there is confusion in the captions, because it should be clearly written that these are bacteria found in the soil from grassland, forest and crop, and not from karst plantation vegetation. Also the title of the manuscript should be changed. Now it's too general. The entire text of the manuscript should be changed to make it clear what kind of soil use is being discussed.
Round 2
Reviewer 2 Report
The current version of the manuscript is much better. They propose to publish the manuscript.